# Synthesis of Carvone Derivatives and In Silico and In Vitro Screening of Anti-Inflammatory Activity in Murine Macrophages

**DOI:** 10.3390/ijms24032263

**Published:** 2023-01-23

**Authors:** Gabriela Moço, Cátia Sousa, Ana Capitão, Stephen Scott MacKinnon, Alcino Jorge Leitão, Alexandrina Ferreira Mendes

**Affiliations:** 1Faculty of Pharmacy, University of Coimbra, 3004-548 Coimbra, Portugal; 2Centre for Neuroscience and Cell Biology, University of Coimbra, 3004-504 Coimbra, Portugal; 3Centre for Innovative Biomedicine and Biotechnology, University of Coimbra, 3004-504 Coimbra, Portugal; 4Interdisciplinary Research Institute, University of Coimbra, 3030-789 Coimbra, Portugal; 5Cyclica Inc., Toronto, ON M5J 1A7, Canada

**Keywords:** chronic inflammation, carvone, chemical synthesis, anti-inflammatory activity, 8- and/or 9-substituted carvone derivatives

## Abstract

The chemical modification of natural compounds is a promising strategy to improve their frequently poor bioavailability and low potency. This study aimed at synthesizing chemical derivatives of carvone, a natural monoterpene with anti-inflammatory properties, which we recently identified, and evaluating their potential anti-inflammatory activity. Fourteen chemical derivatives of carvone were synthesized, purified and their chemical structures confirmed. Noncytotoxic concentrations of the test compounds were selected based on the resazurin reduction assay. Among the tested compounds, four significantly reduced the lipopolysaccharides-induced protein levels of the inducible isoform of the nitric oxide synthase and nitric oxide production and showed a dual effect on pro-IL-1 protein levels in the Raw 264.7 cell line. The Ligand Express drug discovery platform was used to predict the targets of the test compounds, and an enrichment analysis was performed to group the different biological processes and molecular and cellular functions of the tested compounds. Moreover, Ligand Express also predicted that all chemicals evaluated have intestinal and blood–brain barrier permeability, do not inhibit P-gp and do not interact with major receptors. Although presenting anti-inflammatory and some advantageous ADME properties, the tested compounds still have low potency and specificity but may provide novel structures the further chemical modification of which may yield more promising drugs.

## 1. Introduction

Aging is the prevailing risk factor for practically all chronic diseases that limit health span. As the prevalence and incidence of several chronic diseases continue to rise, interventions to increase healthy life expectancy and to delay aging are in high demand [1,2]. The aging process affects all cells, tissues, organs and individuals, causing a decreased ability to maintain homeostasis, especially under stress conditions of both endogenous and exogenous origin. This is generally associated with an increased predisposition to illness and death [3]. Despite being very different, all age-related diseases have in common the occurrence of a low-grade chronic inflammation state that contributes to their morbidity and mortality. This low-grade inflammatory state is associated with an increase in the plasma and tissue levels of several inflammatory mediators, such as chemokines, cytokines and small molecules, including prostaglandins and nitric oxide (NO) [4,5]. Thus, inflammation is emerging as a promising target to counteract aging-related functional decline and associated diseases [6].

Currently available anti-inflammatory drugs, however, are associated with significant adverse effects, especially if used for prolonged periods, thus being inadequate to treat age-related diseases. Natural compounds of plant origin have recently re-emerged due to the fact of their unique structural diversity and wide range of benefits, including anti-inflammatory activities, in a large variety of disease models [7,8,9,10,11,12]. However, natural compounds can present several challenges for therapeutic use, namely, insufficient efficacy, undesirable toxicity, chemical instability, low solubility and poor oral bioavailability [13]. Thus, the chemical optimization of natural active compounds is essential to improve their physicochemical properties and/or to increase their potency and safety, thus yielding suitable leads. This is especially important for monoterpene compounds whose volatility is a major drawback that significantly limits their use as active ingredients for the large-scale production of medicines [14]. In a previous study of monoterpenes with anti-inflammatory activity, we screened compounds bearing the *p*-menthane nucleus and identified carvone (C_10_H_14_O), a monoterpene ketone that exists as two different enantiomers, S-(+)- and R-(−)-carvone, as the most potent [14]. Nonetheless, both carvone enantiomers showed low potency, in the hundred micromolar range, regarding NO production, which precludes their potential therapeutic use. Moreover, the low molecular weight, high volatility and low water solubility [15] of carvone enantiomers have a negative impact on their drug-likeness. Thus, this study aimed at improving those characteristics of the carvone enantiomers by chemical modification. Taking advantage of our previous SAR study in which we found that the existence of the double bond conjugated to the carbonyl group, on the ring structure, is an especially important feature for anti-inflammatory activity [14], while the presence of the isopropenyl group seemed to be less relevant, we used two different molecular modification approaches to synthesize S-(+)- and R-(−)-carvone derivatives.

Both approaches were supported on the reactivity of the double bond of the isopropenyl group. The addition of a hydroxyl group permits the formation of several ether and ester derivatives, while the epoxidation provides a reactive intermediate group that can be opened by nucleophiles in mild conditions (Figure 1).

We obtained and purified fourteen chemical derivatives of the carvone enantiomers and confirmed their chemical structures. Four compounds were obtained in sufficient amounts to be tested for anti-inflammatory effects and cytotoxicity. For that, we treated the mouse macrophage cell line, Raw 264.7, with bacterial lipopolysaccharides (LPS), a widely used cell model of inflammation [16,17,18], and evaluated the ability of the test compounds to inhibit the expression and activity of inflammatory mediators, namely, the inducible isoform of NO synthase (iNOS), which produces large amounts of NO, and the pro-inflammatory cytokine, interleukin-1β (IL-1β). In parallel, the Ligand Express drug discovery platform was used to predict ADME and identify potential molecular targets of the compounds synthesized, as well as of 8-hydroxycarvotanacetone, which we previously found to maintain anti-inflammatory properties [14] and was produced as an intermediary in the synthesis of the other carvone derivatives. Since we found previously that the anti-inflammatory activity of S-(+)-carvone results, at least in part, from its ability to increase the activity of Sirtuin-1 (SIRT1) [19], a NAD^+^-dependent deacetylase that plays crucial roles in regulating metabolism and stress responses [20,21], we also used Ligand Express to predict the affinity of the compounds synthesized to SIRT1 and the other SIRT family members.

## 2. Results

### 2.1. Synthesis and Structural Confirmation of the Carvone Derivatives

Figure 1 shows the structures of the derivatives that were synthesized starting from R-(−)- or S-(+)-carvone and their respective synthesis processes.

The formation of alcohols by the acid-catalyzed addition of water to alkenes is a common reaction in organic chemistry and involves a carbocation intermediate. The alkene is protonated, then reacted with water. This mechanism explains the formation of the more highly substituted alcohol from asymmetrical alkenes, following Markovnikov’s rule [22]. Nevertheless, this reaction is reversible and, as the reverse reaction is also acid-catalyzed, the final yields are usually poor. Furthermore, in the case of carvone, a rearrangement of tertiary carbocation can occur and, via aromatization of the ring, the more stable carvacrol was also obtained as a byproduct.

R-(−) and S-(+)-8-hydroxycarvotacetone (**3** and **4**) were prepared by hydration reaction [23], stirring the R-(−)- or S-(+)-carvone with a 50% aqueous solution of sulfuric acid, at 0 °C for 24 h. Extraction with hexane:ether (3:1) removed the byproduct and the remaining carvone, and further extraction of the aqueous layer with ether (for 24 h) and ethyl acetate, followed by purification with flash chromatography, afforded the pure compound **3** (or **4**) in less than a 50% yield. The identities of compounds **3** and **4** were easily checked using the analytical data (see Section 4 and the Appendix A). Especially, the ^1^H NMR singlets at 1.23 and 1.24 ppm, from the two methyl groups of the isopropyl group, and the peak at 3432 cm^−1^ in the FT-IR spectrum were characteristic of the O–H stretch that marked the difference between the synthesized compounds and the start product, carvone. In contrast, the doublet at 6.76 ppm (^1^H NMR), the peak at 200.48 ppm (^13^C NMR) and the signal at 1650 cm^−1^ (FT-IR) show that the α, β-unsaturated ketone group of the carvone remained unaltered (NMR data of carvone is presented in Appendix A). Since **3** and **4** were enantiomers, all of the analytical data, except the [α]_D_, were the same.

The conversion of alcohols **3** and **4** into esters was performed by O-acylation of the tertiary alcohol group with acetic anhydride in the basic catalysis (the use of 4-dimethylaminopyridine instead of pyridine has been recognized to be more effective as a catalyst for the acylation of secondary or tertiary alcohols in the presence of acid anhydride [24]). The reaction occurs overnight at room temperature and usually affords good chemical yields. R-(−)-8-acetoxycarvotanacetone (**7**) and S-(+)-8-acetoxycarvotanacetone (**8**) were obtained in a 71% and 35% yield, respectively. The analytical data of the compounds were clear (cfr. Section 4 and Appendix A). For instance, the absence of the O–H stretch peak and the appearance of the peak of the ester group at 1727 cm^−1^ (C=O stretch) confirmed the reaction. Since **7** and **8** were enantiomers, all of the analytical data, except the [α]_D_, were the same.

Another attempt to obtain derivatives of carvone was via epoxidation. Epoxides are extremely important synthetic intermediates due to the high reactivity of the strained oxirane ring. Carvone has two double bonds, but prior studies [14] have shown that the enone group is important for the biological activity and should be preserved. The external double bond can be selectively epoxidized [25] using peroxyacids, such as meta-chloroperoxybenzoic acid (m-CPBA) or magnesium monoperoxyphthalate (MMPP), while the electron-deficient α,β-unsaturated ketone remained unaltered. The epoxidation of R-(−)-carvone with m-CPBA resulted in compound **5** in a 66% yield, while, by the same method, 59% of compound **6** was obtained. The identities of the epoxides **5** and **6** were easily confirmed by the analytical data (cfr. Section 4 and Appendix A). Especially, the peak at 6.75 ppm (^1^H NMR), the peak at 198.69 ppm (^13^C NMR) and the signal at 1668 cm^−1^ (FT-IR) show that the α,β-unsaturated ketone group of carvone remained unaltered. On the other hand, we observed the disappearance of the signals that correspond to the methylene groups of carvone, which confirms the epoxidation reaction. R-(−)- or S-(+)-8-epoxycarvotanacetone, were obtained in a nonstereoselective manner, using m-CPBA in dichloromethane. In future studies, stereoselectivity could be achieved using an enantioselective system, such as the tert-butyl hydroperoxide (TBHP) in the presence of an asymmetric catalyst (Sharpless epoxidation or Jacobsen–Katsuki epoxidation) [26].

Epoxides can be further opened by nucleophiles [27] in basic or acid catalysis. The ring opening of epoxides with nucleophilic reagents is a very powerful tool for the preparation of various 1,2-disubstituted compounds, such as β-alkoxy alcohols [28]. As referred, the epoxidation reaction was not stereoselective. The mixture of the epimers was not separated, and the epoxides were opened in mild conditions, with alcohols using catalytic amounts of hydrazine sulfate [29].

Despite the expected regioselectivity of the process (Markovnikov’s rule), when R-(−)-8-epoxycarvotanacetone was dissolved in methanol in the presence of catalytic amounts of hydrazine sulphate, the compounds **9** and **10**, were obtained only in a 1:1.3 ratio of regioisomers. Under the same conditions, S-(+)-8-epoxycarvotanacetone afforded the compounds **11** and **12**. The ratio of regioisomers was equivalent. The poor yields of the process (below 50%) could be optimized using an increased amount of substrate. As the objective was to prepare the epoxide in a stereoselective manner, the processes were not yet optimized. As for the prior compounds, the analytical data show evidence of the obtention of the desired β-methoxy alcohols (cfr. Section 4 and Appendix A). The peak above 3400 cm^−1^ in all FT-IR spectra, characteristic of the O–H stretch, proves the opening of the oxirane ring. Since the epoxidation with m-CPBA was not stereoselective, compounds **9**, **10**, **11** and **12** were obtained as a mixture of diastereoisomers, and most of the signals of the NMR spectra duplicated.

The same method, using ethanol instead of methanol, was also performed. However, the purification of the ethoxy alcohols **13** and **14**, from R-(−)-8-epoxycarvotanacetone, and **15** and **16** from S-(+)-8-epoxycarvotanacetone was a challenging process. The NMR spectra of the compounds showed the obtention of the desired β-ethoxy alcohols but with poor degree of purity (cfr. Section 4 and Appendix A). This reaction follows the same steps as the previous ones, except for the use of 5 mL of ethanol instead of methanol.

### 2.2. Cell Viability of Raw 264.7 Macrophages Treated with the Test Compounds

The compounds framed by rectangles in Figure 1, –S-(+)-8-hydroxycarvotanacetone or S-(+)-8-hydroxycarvone (**4**), R-(−)-8-acetoxycarvotanacetone or R-(−)-8-acetoxycarvotanacetone (**7**), S-(+)-8-acetoxycarvotanacetone or S-(+)-8-acetoxycarvone (**8**) and (5S)-8-hydroxy-9-methoxycarvotanacetone or (5S)-8-hydroxy-9-methoxycarvone (**11**), were obtained in sufficient amounts to proceed to the in vitro evaluation of cytotoxicity and anti-inflammatory activity. Since we previously evaluated the cytotoxicity of S-(+)-8-hydroxycarvone (**4**) and its ability to inhibit LPS-induced NO production, this compound was only tested for the evaluation of iNOS and pro-IL-1β protein levels.

The results in Figure 1 show that in the LPS-treated cells, none of the test compounds (**7**, **8** and **11**) had cytotoxic effects at the concentrations tested. The highest concentration of compound **8** (2400 μM) was not completely miscible with the aqueous medium due to the fact of its lipophilicity, and thus, it was not tested. Although results for some concentrations showed statistically significant differences relative to the cells treated with LPS alone, the effect sizes were very small and, thus, biologically irrelevant. On the other hand, in the absence of LPS, the absorbance intensity in the cells treated with the test compounds decreased to variable extents, although only the highest concentration of (5S)-8-hydroxy-9-methoxycarvone (**11**) caused a decrease higher than 30% relative to the control cells, which, thus, can be considered as representing cell toxicity. Nonetheless, the effect size was small, with the remaining metabolic activity achieving 57.80% of that in the control cells.

### 2.3. The Test Compounds Reduced LPS-Induced NO Production in Raw 264.7 Macrophages

At the highest concentrations tested (1200 and/or 2400 μM), none of the test compounds affected the basal NO production when added to macrophage cultures in the absence of LPS (Appendix A), while all significantly reduced LPS-induced NO production in a concentration-dependent manner (Figure 2). The potency of the test compounds was determined mathematically from the concentration–response curves, shown in Figure 2, by calculating the concentration required to inhibit NO production by 50% (IC_50_). The lowest IC_50_ value (436.5 μM) was obtained with S-(+)-8-acetoxycarvone (**8**), followed by its enantiomer R-(−)-8-acetoxycarvone (521.8 μM, **7**) and by (5S)-8-hydroxy-9-methoxycarvone (1010 μM, **11**).

### 2.4. Effect of the Test Compounds on LPS-Induced iNOS and IL-1β Expression in Raw 264.7 Macrophages

To further confirm the anti-inflammatory properties of the test compounds, iNOS and IL-1β protein levels were evaluated by Western blotting. All test compounds significantly decreased the LPS-induced iNOS protein levels in Raw 264.7 macrophages, although at different concentrations (Figure 3). Despite the inter-assay variability that prevented statistically significant results to be obtained with some concentrations of each test compound, the results clearly show that compounds **7** and **8** were the most effective in inhibiting LPS-induced iNOS expression, in agreement with the IC_50_ values obtained for the inhibition of NO production. IL-1β is produced as a precursor form, pro-IL-1β, which is partially cleaved before being secreted. Pro-IL-1β is cleaved by caspase-1, which is activated by a multiprotein complex, termed inflammasome, in response to a variety of inflammatory stimuli [30]. Thus, even when the IL-1β expression is strongly induced, as in response to LPS, its intracellular levels are usually low or undetectable, and only the unprocessed form with a molecular weight of approximately 30 kDa can be detected, for instance, by Western blotting. The antibody used in this study recognizes an amino acid sequence that is present in both the immature and mature IL-1β forms. Unexpectedly, only the highest concentration of R-(−)-8-acetoxycarvone (**7**) decreased pro-IL-β levels significantly relative to cells treated with LPS alone (Figure 4). On the contrary, the other test compounds showed a tendency to increase the pro-IL-1β levels. Even though statistical significance was not achieved, two out of the three independent assays performed with each compound showed large increases in pro-IL-1β levels compared to cells treated with LPS alone.

### 2.5. Predicted Properties of the Test Compounds

#### 2.5.1. Enrichment Analysis and Target Prediction

Since the number of predicted targets for each compound was very large, we selected only those with relative affinities above 90% for further analysis. For the enrichment analysis, the different biological processes were grouped according to the respective parental term (Figure 5). The results show that carvone was the compound that targets proteins involved in more biological processes (378), followed by 8-acetoxycarvone (281), 8-hydroxy-9-methoxycarvone (51) and 8-hydroxycarvone (25). For all compounds, the parental term with higher representation was “cellular process (c)”, but differences among the compounds were apparent when subdivisions of the parental term, that is, the second parental term, were considered (Figure 6). The second parental term more represented for carvone and 8-acetoxycarvone was “regulation of cellular process (t)”, while “cellular metabolic process (l)” was the most relevant for 8-hydroxycarvone, and for 8-hydroxy-9-methoxycarvone it was “cellular component organization or biogenesis (i)”. Considering all of the cellular processes affected by each compound, it was possible to identify different patterns. Carvone and 8-acetoxycarvone affected a large number of similar processes (16 and 17, respectively), while 8-hydroxycarvone affected only five, and 8-hydroxy-9-methoxycarvone affected an intermediate number of processes (13) of which the top two (i and l) were the same more significantly affected by 8-hydroxycarvone.

Regarding the molecular functions of the targeted processes, the enrichment analyses demonstrated that the 8-acetoxycarvone targets were involved in more molecular functions (97) than those targeted by carvone (64), 8-hydroxy-9-metoxycarvone (39) and 8-hydroxycarvone (36), in agreement with the observations relative to the numbers of biological and cellular processes affected by each one. Considering the parental terms corresponding to the different molecular functions (Figure 7), carvone showed a higher probability of modulation of protein binding, followed by oxireductase activity, while the other three compounds affected proteins with similar functions, but with an inverse probability, that is, the proteins affected by these compounds were more probably involved in oxireductase activity, followed by protein binding.

Considering the predicted protein targets with more than a 90% relative affinity for the compounds tested, carvone and the three derivatives studied share 289 common targets, while carvone shared 510 predicted targets with 8-acetoxycarvone, 489 with 8-hydroxy-9-methoxycarvone and 468 with 8-hydroxycarvone (Figure 8). Carvone presented 190 exclusive targets, while 8-hydroxycarvone presented 182; 8-acetoxycarvone presented 170 and 8-hydroxy-9-methoxycarvone presented 157. Considering the individual proteins, T-box transcription factor T (TBXT) and tumor protein p63 (P63) were those for which carvone had the highest relative affinity, with an identical z-score of 2.7, while ETS proto-oncogene 2 (ETS2), a transcription factor, was the highest affinity target of 8-acetoxycarvone, with a z-score of 3.5; membrane-associated guanylate kinase, WW and PDZ domain-containing 1 (MAGI1) had the highest relative affinity (z-score 2.7) for 8-hydroxy-9-metoxycarvone; and dehydrogenase E1 and transketolase domain-containing 1 (DHTKD1) were the highest relative affinity targets of 8-hydroxycarvone (z-score 2.6).

Since we previously identified that SIRT1 is a target of S-(+)-carvone involved in its anti-inflammatory activity [19], we explored the predicted affinity of the four compounds to the various members of the human SIRT family (SIRT1-7). The z-score calculated using DTI prediction model (Figure 9) predicted a higher relative affinity of carvone for SIRT7 (1), followed by SIRTs 1 and 6 (0.8), SIRT3 (0.6) and SIRT5 (0.4). The order of the relative affinities for 8-acetoxycarvoneis SIRTs 3 and 6 (0.4) > SIRT7 (0.1). 8-Hydroxy-9-methoxycarvone was predicted to target only SIRTs 3 and 6 (0.9), while 8-hydroxy-carvone had a low relative affinity for SIRTs 6 (0.5), 5 (0.2) and 3 (0.2). All compounds presented a negative z-score for SIRTs 2 and 4, and only carvone presented a positive z-score for SIRT1.

#### 2.5.2. ADME

Table 1 summarizes the predictions relative to the permeability through the intestine and the blood–brain barrier (BBB) of carvone and the three chemical structures tested, as well as to the probability of the interaction with important proteins whose modulation by drugs can lead to adverse effects.

The results obtained predicted that all four structures were permeable through the intestinal wall and the blood–brain barrier and that none interacted, either as an inhibitor or substrate, with glicoprotein-P (P-gp), an ABC transporter that significantly impairs the oral bioavailability and the SNC distribution of many drugs and constitutes a clinically relevant mechanism of drug–drug interactions [31].

To further validate the predictions offered by Ligand Express, we used two other platforms, SwissADME [32] and pkCSM [33], to predict whether the compounds tested fulfilled the rules of Lipinski, which estimate solubility and permeability. The results presented in Table 2 show the predictions based on the extended rules of Lipinski, provided by the two platforms. All structures, except carvone, comply with all of Lipinski’s rules, excluding the heavy atom score, which was below the minimum for all structures. However, permeability does not require that all criteria are simultaneously fulfilled. Moreover, the other criterium not fulfilled by carvone was molecular weight, which being low, con-tributes to its volatility. The chemical modifications that yielded the compounds tested elicited a sufficient increase in this parameter that not only complied with Lipinski’s rules but apparently was sufficient to decrease the volatility, which was one of the aims of this study.

The predictions in Table 1 also show some disparities in the interaction of the test compounds with various receptors, enzymes and other proteins that can pose particular concerns in terms of toxicity. On the other hand, some of those interactions may eventually have therapeutic interest and may also be investigated. In general, however, most of the proteins included in the analysis are likely not affected by the test compounds. Of notice is the prediction that none of the test compounds will have a positive result in the AMES test for mutagenicity.

## 3. Discussion

The results obtained show that the chemical modification of the isopropenyl group of carvone enantiomers was achievable using common synthetic reactions that resulted in the formation of compounds with the predicted structures (Figure 1). Moreover, and although we could not measure their vapor pressure at room temperature due to the small quantities obtained, the compounds synthesized did not seem to be volatile, which was one of the aims of this study. Nonetheless, the low yield of most compounds did not allow for their evaluation in vitro and also limited the experiments that could be performed with the four compounds tested. Notwithstanding, the results obtained in vitro with the four compounds tested confirm the feasibility of modifying the isopropenyl group maintaining the anti-inflammatory activity, but the low potency of the compounds tested suggests that other modifications of the same group may be more effective. Future studies will be aimed at developing other chemical approaches to improve the yield as well as to synthesize different the derivatives of the carvone enantiomers.

As mentioned above, we plan to perform the epoxidation using an enantioselective system [26] to evaluate the selectivity of the opening reaction and to assess the activity of each diastereomer of the β-alkoxy alcohols. Furthermore, the results obtained with the esters **7** and **8** carried us to the design and synthesis of other derivatives of (R)− and (S)−8-hydroxycarvotanacetone in order to modulate the pharmacokinetic properties and improve the pharmacological activity.

The results of the in vitro assays indicated that compounds **7**, **8** and **11** have anti-inflammatory properties, as evidenced by the decrease in NO production (Figure 2) and iNOS expression (Figure 3), and also confirmed that compound **4** inhibited NO production [14] by inhibiting the LPS-induced iNOS expression (Figure 3). However, the results obtained for IL-1β were less clear, since a tendency for increased amounts of its precursor form were observed (Figure 4). This suggests that the test compounds have an inhibitory effect on the proteolytic processing of pro-IL-1β. Future studies will be directed at evaluating the ability of the test compounds to inhibit the inflammasome and the activity of caspase-1.

Even though only four chemical structures, representing six of the compounds synthesized and carvone, without consideration for enantiomeric forms, were tested for pharmacological effects, access to the Ligand Express drug discovery platform allowed for the prediction of the ADME properties and of affinity for the protein targets relevant for toxicity and for potential therapeutic effects. The results obtained show that all of the tested chemical structures potentially interact with many cellular proteins involved in multiple cellular processes (Figure 5, Figure 6, Figure 7 and Figure 8), which may be undesirable, as it can be a source of secondary, eventually undesirable side effects. Nonetheless, clearly distinct patterns can be observed with the four chemical structures in terms of the cellular pathways potentially affected, highlighting the huge potential for the chemical modification of natural compounds. On the other hand, it is interesting to notice that oxidoreductase activity is one of the molecular functions common to the proteins for which the chemical structures tested show a higher interaction potential, although with different probabilities. Oxidative stress is a major feature of chronic low-grade inflammation and associated diseases [34,35] and is thus a potentially relevant mechanism by which the compounds tested can exert anti-inflammatory effects. Interestingly, carvone is the only compound for which the oxidoreductase activity was not the major function of the potentially targeted proteins, and SIRT1 was a potential target (Figure 9). This agrees with our previous study that showed the ability of S-(+)-carvone to directly interact with and activate SIRT1 [19], thus confirming the validity of the bioinformatic predictions obtained. Nonetheless, the z-score for that interaction was not as high as could be expected from the in vitro results.

In terms of ADME, the predictions offered by Ligand Express agreed with those of the other two platforms, SwissADME and pkCSM, and suggest that the chemical modifications of carvone that yielded the structures tested positively addressed the goals set forth in this study, namely, increasing drug-likeness by decreasing volatility while maintaining intestinal and BBB permeability. Moreover, the lack of inhibition of P-gp by any of the chemical structures tested, as predicted by Ligand Express, is especially relevant as it suggests that the corresponding compounds will have a low probability of causing pharmacokinetic drug–drug interactions at the level of absorption and distribution and will have no obstacles in crossing those barriers. However, the potential of interaction with metabolic enzymes, notably isoforms of the cytochrome P450, was not assessed and, thus, the possibility of the occurrence of pharmacokinetic drug interactions at the level of metabolism is unknown. In terms of potential toxicity, it is relevant that none of the structures were predicted to have a mutagenic potential.

Further studies are required to confirm these predictions, both at the pharmacodynamic and pharmacokinetic levels, but the conjugation of the in vitro results presented and the bioinformatic predictions highlight future directions for research on the compounds tested in order to improve the development of new anti-inflammatory drugs that can efficiently target chronic diseases associated with low-grade inflammation.

## 4. Materials and Methods

### 4.1. Semi-Synthesis of Carvone Derivatives

#### 4.1.1. Chromatography

The reactions were monitored by thin layer chromatography (TLC) using silica gel 60 F_254_ plates with an aluminum support (Merck, Germany). For the column chromatography, we used silica gel 60 (0.063–0.200 mm, Merck, Germany).

#### 4.1.2. Nuclear Magnetic Resonance (NMR)

The NMR spectra were obtained through the Bruker Avance III spectrometer, with TopSpin3 acquisition and processing NMR Software, at 400 MHz (^1^H) or 100 MHz (^13^C), and tetramethylsilane (TMS) was used as the reference. Chemical shifts (δ) are reported in parts per million (ppm), and coupling constants (J) are reported in hertz (Hz).

#### 4.1.3. Infrared Spectroscopy (FT-IR)

The infrared spectra were recorded on a Fourier transform spectrometer, Perkin Elmer Spectrum 400, using ATR and the software PerkinElmer SpectrumIR, Version 10.6.0.

#### 4.1.4. Reagents and Solvents

The solvents and reagents used are commercially available and were used without further purification, except tetrahydrofuran, which was dried by refluxing over calcium hydride for 7 h, distilled from calcium hydride and stored in molecular sieves, and methanol and ethanol which were distilled and stored in activated molecular sieves for, at least, 24 h before use.

#### 4.1.5. Hydration Reaction

R-(−)- and S-(+)-8-hydroxycarvone (**3** and **4**) were prepared by a hydration reaction using a method adapted from [23]. To 500 mg of R-(−)- and S-(+)-carvone at 0 °C, was added slowly 3.3 mL of 50% aqueous sulfuric acid, and the mixture was stirred for 24 h at 0 °C. After extraction with 6 mL of hexane-ether (3:1), the aqueous layer was submitted to extraction with ether (3 × 6 mL for 24 h). The ether solution was washed with brine containing sodium bicarbonate, dried over sodium sulphate anhydrous and evaporated under reduced pressure. The aqueous layer was again submitted to extraction with ethyl acetate (3 × 10 mL for 12 h). The organic phases were washed with brine containing sodium bicarbonate, dried over sodium sulphate anhydrous and evaporated under reduced pressure. The combined crude was purified by flash chromatography (hexane:ethyl acetate 1:0 to 1:1).

Applying the general method of hydration to S-(+)-carvone (**2**), 116 mg (21%) of **4**, viscous liquid was obtained. [α]_D_ = 42° (c. 1.22; CHCl_3_); ^1^H NMR (400 MHz, CDCl_3_): δ = 1.23 (3H, s, –CH_3_); 1.24 (3H, s, –CH_3_); 1.77 (3H, s, –CH_3_C=CH); 2.06–2.60 (5H, m); 6.76 (1H, d, *J* = 6.76 Hz, –C=CH); ^13^C NMR (100 MHz, CDCl3): δ = 27.02 (–CH_3_); 27.31 (–CH_3_); 71.64 (–COH); 200.48 (C=O); FT-IR (ATR): cm^−1^ = 3432 (O–H); 1656 (C=O). This method was also performed with R-(−)-carvone (**1**), affording 122 mg (22%) of **3**, liquid viscous, with the same spectral data. [α]_D_ = −41° (c. 1.43; CHCl_3_).

#### 4.1.6. Esterification of Tertiary Alcohols: Synthesis of R-(−)-8-Acetoxycarvotanacetone (**7**) and S-(+)-8-Acetoxycarvotanacetone (**8**)

This procedure was adapted from [36]. R-(−)- or S-(+)-carvone (124 mg) was dissolved in 4 mL of dry tetrahydrofuran, 0.4 mL of acetic anhydride and 75 mg of 4-dimethylaminopyridine and kept stirring overnight at room temperature. The mixture was successively washed with 1 M solution of acid chloride and brine, dried with anhydrous sodium sulfate, evaporated under reduced pressure and purified by flash chromatography (eluent petroleum ether:ethyl acetate 1:0 to 1:1). R-(−)-8-acetoxycarvotanacetone (**7**) was obtained, after purification by flash chromatography, as a viscous yellow liquid: 88 mg (71% yield, [α]_D_ = −26° (c. 1.41; CHCl_3_).

Using the same procedure with S-(+)-carvone, we obtained 42 mg, (34% yield) of **8**, a viscous yellow liquid. [α]_D_ = 28° (c. 1.22; CHCl_3_); ^1^H NMR (400 MHz, CDCl_3_): δ = 1.48 (6H, s, –CH_3_); 1.78 (3H, s, –CH_3_); 1.98 (3H, s, –OCOCH_3_); 6.74–6.76 (1H, d, *J* = 6.71 Hz, –CH=C); ^13^C NMR (100 MHz, CDCl_3_): δ = 23.30 (–CH_3_); 23.46 (–CH_3_); 44.33 (–OCOCH_3_); 82.88 (–C); 170.37 (C=O); 199.69 (–OCOCH_3_); FT-IR (ATR): cm^−1^ = 1727 (C=O); 1671 (C=O).

#### 4.1.7. Regioselective Epoxidation of Carvone’s Exocyclic Double Bond: Synthesis of (5R)-8-Epoxycarvotanacetone (**5**) and (5S)-8-Epoxycarvotanacetone (**6**)

This procedure was adapted from [37]. Carvone (500 mg) was dissolved in 8 mL of ice-cold dichloromethane and a solution of 75% m-CPBA (85 mg) diluted in 4 mL of dichloromethane, were added dropwise for 10 min. The mixture was stirred at 0 °C (ice bath) for 16 h (TLC control). The mixture was then stirred with 1 mL of 10% aqueous sodium sulfite for 1–2 min, filtered and the solid residue was washed with several portions of dichloromethane. The organic phase was washed with 10% aqueous sodium carbonate (3 × 15 mL) and brine (15 mL), dried with anhydrous sodium sulfate, and the solvent evaporated under reduced pressure. The residue was further purified by flash chromatography (eluent petroleum ether:ethyl acetate 1:0 to 1:1).

When R-(−)-carvone was submitted to the general procedure of epoxidation with m-CPBA, the compound **5** was obtained as a viscous yellow liquid, in 360 mg, 66% yield. ^1^H NMR (400 MHz, CDCl_3_): δ = 1.33 (3H, s, –CH_3_); 1.77 (3H, s, –CH_3_); 2.18–2.30 (5H, m); 2.52–2.60 (2H, m, –CH_2_O); 6.75 (1H, s, –C=CH); ^13^C NMR (100 MHz, CDCl_3_): δ = 18.33 (–CH_3_); 52.36 (–CH_2_O); 57.77 (–C); 198.69 (C=O); FT-IR (ATR): cm^−1^ = 1668 (C=O); 802, 829 (C–O). When the method was performed in S-(+)-carvone, 324 mg, 59% yield of compound **6** was obtained.

#### 4.1.8. Catalytic Ring Opening of the Epoxide with Alcohols as Nucleophiles and Solvent

This procedure was adapted from [29]. Starting with 100 mg of the epoxide **5** or **6**, 5 mL of the appropriate alcohol and 13 mg of hydrazine sulfate salt were added. After approximately 48 h under magnetic stirring at 50 °C (TLC control), the reaction mixture is filtered (the catalyst is recovered) and the filtrate is concentrated under reduced pressure. The resulting residue is purified by flash chromatography (eluent petroleum ether:ethyl acetate 1:0 to 1:1).

Starting with 100 mg of **5**, were obtained 38 mg, 19% yield, for **9** and 50 mg, 25% yield, for **10**, viscous yellow liquids. Analytical data of **10**: ^1^H NMR (400 MHz, CDCl_3_): δ = 1.11 [1,12] (3H, s, –CH_3_); 1.78 (3H, s, –CH_3_C=CH); 2.18–2.57 (5H, m); 3.23 [3.24] (3H, s, –OCH_3_); 3.51–3.59 (2H, m, –CH_2_OH); 6.73–6.75 [6.78–6.79] (1H, d, J1 = 6.0 Hz [J2 = 5.9 Hz], –C=CH); ^13^C NMR (100 MHz, CDCl_3_): δ = 16.55 [16.72] (–CH_3_); 49.49 [49.57] (–OCH_3_); 64.13 [64.37] (–CH_2_OH); 77.70 [77.74] (–C); 199.86 [200.23] (C=O); FT-IR (ATR): cm^−1^ = 3454 (O–H); 1661 (C=O); 1061 (C–O). The values in square brackets represent the data for the compound’s diastereoisomer.

Starting from 100 mg of **6**, we obtained two viscous yellow liquids, 42 mg, 21% yield of **11** and 55 mg, 28% yield of **12**. Analytical data of **11**: ^1^H NMR (400 MHz, CDCl_3_): δ = 1.13 [1.15] (3H, s, –CH_3_); 1.77 (3H, s, –CH_3_=CH); 2.22–2.31 (5H, m); 3.20–3.25 (2H, m, –CH_2_OH); 3.38 [3,38] (3H, s, –OCH_3_); 6.73 [6.79] (1H, d, J1 = 5.8 Hz [J2 = 5.8 Hz], –CH); ^13^C NMR (100 MHz, CDCl_3_): δ = 20.83 [21.44] (–CH_3_); 59.39 [59.43] (–OCH_3_); 72.72 [72.88] (–C); 78.13 [78.13] (–CH_2_OMe); 199.12 [200.41] (C=O); FT-IR (ATR): cm^−1^ = 3460 (O–H); 1662 (C=O); 1107 (C–O). The values in the square brackets represent the data for the diastereoisomer.

Since the compounds **9**, **10**, **11** and **12** were obtained as a mixture of diastereoisomers, most of the signals of the NMR spectra became duplicated.

The same method, using ethanol instead of methanol, was also performed. However, the purification of the ethoxy alcohols **13** and **14**, from R-(−)-8-epoxycarvotanacetone, and **15** and **16**, from S-(+)-8-epoxycarvotanacetone, were a challenging process.

Starting from 100 mg of **5**, we were able to obtain two viscous yellow liquids, 16 mg, 8% yield, for **13** and 21 mg, 11% yield, for **14**.

Starting from 100 mg of **6**, we obtained two viscous yellow liquids, in 22 mg, 10% yield, for **15** and 33 mg, 15% yield, for **16**. Analytical data of **15** (impure, selected peaks): ^1^H NMR (400 MHz, CDCl_3_): δ = 1.00 (3H, s, –CH_3_); 1.42 (3H, s, –CH_3_C=CH); 4.15–417 (2H, m, –CH_2_OEt); 6.72–6.74 (1H, m, –CH=C); ^13^C NMR (100 MHz, CDCl_3_): δ = 65.0 (–CH_2_OEt); 72.5 (–COH); 135.7 (CH=C); 145.4 (CH=C); 203.6 (C=O); FT-IR (ATR): cm^−1^ = 3459 (O–H); 1671 (C=O); 1061 (C–O). Analytical data of **16**: ^1^H NMR (400 MHz, CDCl_3_): δ = 1.12 [1.13] (3H, s, –CH_3_); 1.18 [1.18] (3H, t (*J* = 6.0 Hz), –OCH_2_CH_3_); 1.77 (3H, s, –CH_3_C=CH); 2.21–2.26 (5H, m); 3.39–3.45 (2H, m, –OCH_2_CH_3_); 3.54 [3.54] (2H, s, –CH_2_OH); 6.74 [6.78] (1H, d, J1 = 6.0 Hz [J2 = 5.9 Hz], –CH=C); ^13^C NMR (100 MHz, CDCl_3_): δ = 15.75 [16.04] (–CH_3_CH_2_O); 17.10 [17.49] (–CH_3_); 56.84 [56.92] (–OCH_2_CH_3_); 64.62 [64.99] (–CH_2_OH); 77.60 [77.60] (–C); 199.97 [200.36] (C=O); FT-IR (ATR): cm^−1^ = 3488 (O–H); 1671 (C=O); 1060 (C–O). The values in the square brackets represent the data for the diastereoisomer.

### 4.2. Cell Culture and Treatments

The Raw 264.7 cells were plated at a density of 3 × 10^5^ cells/mL and cultured in DMEM (Gibco, Billings, MT, USA) supplemented with 10% non-heat inactivated fetal bovine serum (FBS; Gibco, USA), 100 U/mL penicillin (Sigma-Aldrich Co., St Louis, MO, USA) and 100 μg/mL streptomycin (Sigma-Aldrich Co.) for 24 h before the experiments. For the cell treatments, the test compounds were dissolved in dimethyl sulfoxide (DMSO; Honeywell, Germany), and LPS from *Escherichia coli* 026:B6 (Sigma-Aldrich Co.) was dissolved in phosphate-buffered saline (PBS). DMSO was used as a vehicle and added to the positive control (untreated cells + DMSO) as well as to LPS-treated cells to match the same concentration as in the cells treated with the test compounds. The final concentration of DMSO was 0.1% (*v/v*). The test compounds or DMSO were added to the Raw 264.7 cells 1 h before the pro-inflammatory stimulus, 1 μg/mL LPS, and maintained for the rest of the experimental period (18 h) at 37 °C.

### 4.3. Selection of the Noncytotoxic Concentrations of the Test Compounds by the Resazurin Reduction Assay

To select the noncytotoxic concentrations of the test compounds, the resazurin reduction assay was used. Resazurin is a redox dye used as an indicator of active metabolism with several applications, including cell viability, proliferation and toxicity studies. This assay is based on the intracellular reduction of resazurin—a nonfluorescent compound—to resorufin, a fluorescent and pink-colored compound, by mitochondrial or microsomal enzymes that use NADH or NADPH as electron sources. Only living, metabolically active cells can reduce resazurin; thus, the fluorescence and absorbance intensity are directly proportional to the number of viable cells [38]. Comparison of the fluorescence or absorbance intensity obtained in the test condition with that observed in the control indicates whether the test condition affects the number of metabolically active cells, reflecting the cell viability and/or proliferation. The resazurin solution was added to each well to a final concentration of 50 μM, 90 min before the end of the 18 h treatment. Then, absorbances at 570 nm (test wavelength) and 620 nm (reference wavelength) were read in a Biotek Synergy HT plate reader (Biotek, Winooski, VT, USA). According to the standard for cytotoxicity assessment, ISO 10993-5, cell toxicity is defined as a decrease in the metabolically active cells larger than 30% relative to the control condition. Thus, this standard was used to select noncytotoxic concentrations of the test compounds.

### 4.4. Nitric Oxide Production

The NO production was measured based on the amount of nitrite accumulated in the culture supernatants using the Griess assay. This consists of a two-step diazotization reaction in which the NO derived nitrosating agent, dinitrogen trioxide (N_2_O_3_), generated from the acid-catalyzed formation of nitrous acid from nitrite (or autoxidation of NO), reacts with sulfanilamide to produce a diazonium ion, which is then coupled to N-(1-napthyl)ethylenediamine to form a chromophoric azo product that absorbs strongly at 540 nm [39]. Equal volumes of the culture supernatants and reagents (equal volumes of 1% (*w*/*v*) sulphanilamide in 5% (*v*/*v*) phosphoric acid and 0.1% (*w*/*v*) N-(1-napthtyl) ethylenediamine dihydrochloride) were mixed and incubated for 10 min at room temperature and placed in the dark. The concentration of accumulated nitrite in the culture supernatants was calculated by interpolation of the absorbance of each sample, read in a Biotek Synergy HT plate reader (Biotek) at 550 nm, in a standard curve of sodium nitrite.

### 4.5. Western Blotting

The total cell protein extracts were prepared in RIPA buffer (150 mM sodium chloride, 50 mMTris, 5 m methylene glycol-bis(2-aminoethylether)-N,N,N′,N′-tetraacetic acid, 0.5% sodium deoxycholate, 0.1% sodium dodecyl sulfate and 1% Triton X-100), supplemented with protease (Complete, Mini, Roche Diagnostics, Mannheim, Germany) and phosphatase (PhosSTOP, Roche Diagnostics, Mannheim, Germany) inhibitors. Each sample (25 μg protein) was denatured at 95 °C in sample buffer (5% SDS, 0.125 M Tris–HCl, pH 6.8, 20% glycerol, 10% 2-mercaptoethanol and bromophenol blue) for 5 min and separated by SDS-PAGE under reducing conditions. The proteins were electro-transferred onto PVDF membranes using a wet transfer system at 350 mA for 210 min. After blocking with 5% nonfat dry milk in Tris-buffered saline (TBS)-Tween 20 (0.1%) for 2 h, followed by five 5 min washes in TBS-Tween 20, the membranes were probed overnight at 4 °C with a rabbit monoclonal anti-interleukin-1β antibody (1:1000 dilution, Abcam, Cambridge, UK) or with a mouse monoclonal anti-iNOS antibody (1:500 dilution, R&D Systems, Minneapolis, MN, USA). Then, the membranes were washed again 5 times in TBS-Tween 20 and incubated with anti-rabbit or anti-mouse secondary antibodies, diluted at 1:5000 (Santa Cruz Biotechnology, Dallas, TX, USA), for 1 h. β-Tubulin was detected with a mouse monoclonal antibody (1:20,000 dilution, Sigma-Aldrich Co.) and was used as the loading control. The immune complexes were detected with Clarity Western ECL Substrate (Bio-Rad Laboratories, Inc., Hercules, CA, USA) using the imaging system ImageQuantTM LAS 500 (GE Healthcare). The image analysis was performed with TotalLab TL120 1D V2009 software (Nonlinear Dynamics Ltd., Newcastle upon Tyne, UK).

### 4.6. ADME, Toxicity and Target Prediction Using Ligand Express

Three different structures, 8-hydroxycarvotanacetone, 8-acetoxycarvotanacetone and 8-hydroxy-9-methoxycarvotanacetone, were tested using the Ligand Express platform. To further validate the results, two other publicly accessible platforms, SwissADME and pkCSM, were used to predict the compliance of the test structures with the Lipinski criteria for permeability and solubility which predict drug-likeness.

#### 4.6.1. ADME Properties

The ADME proprieties of the compounds identified above and carvone were predicted in Ligand Express using prebuilt pareto-optimal embedded modeling (POEM) release models [40].

#### 4.6.2. Drug–Target Interaction (DTi) Prediction

The target proteins for the three compounds and carvone were predicted with the MatchMaker DTi prediction model [41] in Ligand Express. MatchMaker screens and ranks 8624 human proteins for complementarity. The ranking is calculated as:$$\left(1-\frac{rank}{total\;number\;of\;proteins}\right)*100$$

The top ranking targets were used to perform a gene ontology (GO) enrichment analysis (rank > 90).

A Venn diagram to compare the obtained targets (rank > 90) for 8-acetoxycarvotanacetone, 8-hydroxy-9-methoxycarvotanacetone, 8-hydroxycarvotanacetone and carvone was performed using the ggvenn package [42] in RStudio (version 4.1.1) [43].

The affinity of the different compounds for human SIRTs (1–7) were evaluated through the z-score and plotted in a heatmap using the heatmap function in RStudio (version 4.1.1) [43].

#### 4.6.3. Enrichment Analysis

The gene ontology enrichment analysis (biological processes, cellular components and molecular functions) for the predicted targets with a rank > 90 was performed using the clusterProfiler package [44] in RStudio (version 4.1.1) [43], with a cutoff criterion of a *p* value < 0.05. Parental nodes were retrieved using the GOfuncR package [45] in RStudio (version 4.1.1). The results with more than 1% of expression were visualized using meta-chart [46].

### 4.7. Statistical Analysis of the In Vitro Results

The results are presented as the means ± SEM. The statistical analysis was performed using GraphPad Prism version 8.0.2 (GraphPad Software, San Diego, CA, USA). The statistical significance was evaluated with one-way ANOVA with the Dunnet post-test for comparison of multiple conditions or with the *t*-test to compare two distinct conditions. The results were considered statistically significant at *p* < 0.05.

## Data Availability

Data generated in this study are available from the corresponding author upon reasonable request.

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
