# Peer review of "Synthesis of Carvone Derivatives and In Silico and In Vitro Screening of Anti-Inflammatory Activity in Murine Macrophages"

_ijms, 2023, doi:10.3390/ijms24032263_

Round 1

Reviewer 1 Report

-The authors give references of their previous work in abstract which is not required in abstract. It should be mentioned in last paragraph of introduction.

-In results section, authors provide complete analytical data of compounds which should be provided in experimental section. Results section should be revised accordingly.

-Why did the authors choose simple anhydrides since there are other heterocyclic halides available? In general, heterocycles tend to effect more on biological activities.

-How did the authors check the stereoselectivity/regioselectivity in case of epoxide ring opening products? Ratio of both the regio-isomers/diastereomers should be mentioned. Similarly, NMR data for both the major and minor isomers should be clearly mentioned.

- Did authors observe any nucleophilic ring opening product by hydrazine?

-Figure 1 should be renamed as scheme 1 in which yields of the synthesized derivatives should be mentioned.

-In supplementary files, NMR spectra should be expanded and each peak other than impurities should be integrated.

-Lines 159-167 needs references being general statements. Appropriate references may be cited.

-Lines 179-181 need some references. Appropriate references may be cited.

-English language needs to be improved. Some spelling mistakes (for example, in line 13 'synthetizing' should be replaced with 'synthesizing'), grammatical errors have been observed. The authors need to carefully check the manuscript.

Author Response

Reviewer 1

-The authors give references of their previous work in abstract which is not required in abstract. It should be mentioned in last paragraph of introduction.

R: The reference in the abstract was removed. It was already mentioned in other sections of the manuscript, including the discussion.

-In results section, authors provide complete analytical data of compounds which should be provided in experimental section. Results section should be revised accordingly.

R: As suggested by reviewers 1 and 2, details of synthesis reactions were included in the methods section. The results section was revised to include only information regarding the characterization of the compounds obtained in the various reactions.

-Why did the authors choose simple anhydrides since there are other heterocyclic halides available? In general, heterocycles tend to effect more on biological activities.

R: In the current study, we used common reagents to alter the –OH group and consequently, the properties of the molecule. We are planning to synthesize and test other derivatives (namely, ester and ether) in a near future and, certainly, the use of heterocyclic halides is an excellent idea.

-How did the authors check the stereoselectivity/regioselectivity in case of epoxide ring opening products? Ratio of both the regio-isomers/diastereomers should be mentioned. Similarly, NMR data for both the major and minor isomers should be clearly mentioned.

R: We used m-CPBA in the epoxidation reaction, so the epoxide was obtained as a mixture of diastereomers. NMR spectra didn´t give us information about the ratio of diastereomers. In addition, no stereoselectivity studies were performed for the opening reaction.

Concerning the regioselectivity, the compounds were isolated by flash chromatography and poor regioselectivity (1.3:1) was observed. As expected (ref 23), the 9-methoxy derivative was the main regioisomer.

In future studies, we plan to perform the asymmetric epoxidation of the isomers of carvone and then, study the stereoselectivity and regioseletivity of the epoxy opening reaction.

- Did authors observe any nucleophilic ring opening product by hydrazine?

R: No, we didn’t as the reaction conditions were very mild and so unlikely to allow the nucleophilic attack at the ring.

-Figure 1 should be renamed as scheme 1 in which yields of the synthesized derivatives should be mentioned.

R: Figure 1 was changed to Scheme 1 and information regarding yields of the derivatives synthesized and reagents used was added.

-In supplementary files, NMR spectra should be expanded and each peak other than impurities should be integrated.

R: The spectra were changed as suggested. Moreover, we also replaced the carbon spectra to better show all peaks.

-Lines 159-167 needs references being general statements. Appropriate references may be cited.

R: We added appropriate references, as suggested.

-Lines 179-181 need some references. Appropriate references may be cited.

R: We added appropriate references, as suggested.

-English language needs to be improved. Some spelling mistakes (for example, in line 13 'synthetizing' should be replaced with 'synthesizing'), grammatical errors have been observed. The authors need to carefully check the manuscript.

R: We apologize and thank the reviewer for the careful and thorough revision. All the text was checked for spelling and grammar errors and several changes were made to correct those mistakes and improve readability.

Reviewer 2 Report

Taking natural products as the leader and finding new active leaders through structural optimization is one of the important ways to develop new drugs. This study aimed at synthesizing chemical derivatives of carvone and evaluating their potential anti-inflammatory activity. Fourteen chemical derivatives of carvone were synthesized, purified, and their chemical structures were confirmed. The research showed that four significantly reduced lipopolysaccharides induced the inducible isoform of the nitric oxide synthase protein levels and nitric oxide production and showed a dual effect on pro-IL-1 protein levels in Raw 264.7 cell line. The Ligand Express drug discovery platform was used to predict the targets of the test compounds and enrichment analysis. In general, this part work is the result of further in-depth study of their previous work (Biomedicines 2021, 9, 777). However, there are still several issues to be solved be addressed.

1. It is suggested to adjust the data 1H NMR of compounds to the section of Materials and Methods. In the section on Results, the references cited in the synthesis process should be added.

2. The characterization data of compounds 13 or 15 should also be provided.

3. The purity of the compound will directly affect its activity. The NMR spectra obviously have impurity peaks (such as Figure S2 and Figure S17.), which will inevitably affect the accuracy of the results.

4. In the 1H NMR spectra of the target compounds, the hydrogen spectrum integration is incomplete.

5. Whether the control drug is required for the cell viability test.

Author Response

Reviewer 2

Taking natural products as the leader and finding new active leaders through structural optimization is one of the important ways to develop new drugs. This study aimed at synthesizing chemical derivatives of carvone and evaluating their potential anti-inflammatory activity. Fourteen chemical derivatives of carvone were synthesized, purified, and their chemical structures were confirmed. The research showed that four significantly reduced lipopolysaccharides induced the inducible isoform of the nitric oxide synthase protein levels and nitric oxide production and showed a dual effect on pro-IL-1 protein levels in Raw 264.7 cell line. The Ligand Express drug discovery platform was used to predict the targets of the test compounds and enrichment analysis. In general, this part work is the result of further in-depth study of their previous work (Biomedicines 2021, 9, 777). However, there are still several issues to be solved be addressed.

  1. It is suggested to adjust the data 1H NMR of compounds to the section of Materials and Methods. In the section on Results, the references cited in the synthesis process should be added.

R: As suggested by reviewers 1 and 2, details of synthesis reactions were included in the methods section. The results section was revised to include only information regarding the characterization of the compounds obtained in the various reactions. Relevant references were added as suggested.

  1. The characterization data of compounds 13 or 15 should also be provided.

R: The NMR (1H, 13C and DEPT) and FT-IR spectra (S17, S18 and S19) of compound 15 were added to the Supplementary Materials document and the characterization data was added to section 4.1.8 of Materials and Methods, as requested.

  1. The purity of the compound will directly affect its activity. The NMR spectra obviously have impurity peaks (such as Figure S2 and Figure S17.), which will inevitably affect the accuracy of the results.
  2. In the 1H NMR spectra of the target compounds, the hydrogen spectrum integration is incomplete.

R: The NMR spectra were revised to show all the relevant information, as suggested by the reviewer.

  1. Whether the control drug is required for the cell viability test.

R: We didn’t use any control drug in the cytotoxicity assay because we were not looking for compounds that would affect cell viability or proliferation. The single purpose of the assay was to ensure that eventual inhibitory effects were not due to cell death which is clearly shown by the significant anti-inflammatory effect and lack of significant loss of cell viability. Moreover, the method used is very well established at our laboratory and we used it to test dozens of different compounds with varying toxicities (for examples besides those cited in the manuscript, please see: doi: 10.1055/s-0029-1186085; doi: 10.3109/13880209.2014.970701; doi: 10.1016/j.ejphar.2015.01.018).

Reviewer 3 Report

The manuscript entitled “Synthesis of Carvone Derivatives and In Silico and In Vitro Screening of Anti-Inflammatory Activity in Murine Macrophages” by Moco et al. synthesized and purified fourteen chemical derivatives of carvone and evaluated its potent anti-inflammatory activities. Results showed that four compounds significantly reduce lipopolysaccharides-induced the inducible isoform of the nitric oxide synthase protein levels and nitric oxide production. Further authors have predicted the ADMET properties of 3 compounds. I have a few suggestions/queries that need to be addressed prior to acceptance of the manuscript. I recommend a minor revision of the manuscript. Here are my suggestions for the manuscript:-

1.    In section 4.5 Western Blotting, author should elaborate the method, like time period for washing of blot.

2.    Manuscript need to be screen for typos and grammatical mistakes.

3.    Authors should use a reliable tool like pkCSM to predict the ADMET properties. The mentioned tool will predict the other properties like water solubility, BBB,CNS permeability, hepatoxicity, and skin sensitivity of compounds also.

4.    Author should predict the Lipinski properties also along with ADMET properties.

Author Response

Reviewer 3

The manuscript entitled “Synthesis of Carvone Derivatives and In Silico and In Vitro Screening of Anti-Inflammatory Activity in Murine Macrophages” by Moco et al. synthesized and purified fourteen chemical derivatives of carvone and evaluated its potent anti-inflammatory activities. Results showed that four compounds significantly reduce lipopolysaccharides-induced the inducible isoform of the nitric oxide synthase protein levels and nitric oxide production. Further authors have predicted the ADMET properties of 3 compounds. I have a few suggestions/queries that need to be addressed prior to acceptance of the manuscript. I recommend a minor revision of the manuscript. Here are my suggestions for the manuscript:

  1. In section 4.5 Western Blotting, author should elaborate the method, like time period for washing of blot.

R: Further details were added to section 4.5, as suggested.

  1. Manuscript need to be screen for typos and grammatical mistakes.

R: We apologize and thank the reviewer for the careful and thorough revision. All the text was checked for spelling and grammar errors and several changes were made to correct those mistakes and improve readability.

  1. Authors should use a reliable tool like pkCSM to predict the ADMET properties. The mentioned tool will predict the other properties like water solubility, BBB,CNS permeability, hepatoxicity, and skin sensitivity of compounds also.
  2. Author should predict the Lipinski properties also along with ADMET properties.

R: The Lipinski properties predicted with SwissADME and pkCSM were included in table 2 and in the results section 2.5.2. The discussion and methods sections were also modified accordingly.

Round 2

Reviewer 1 Report

The authors have now revised the manuscript and is now acceptable for publication.

Reviewer 2 Report

The authors have made careful modifications according to the review comments, and it is recommended to accept it directly.